# Influence of the Flexible Tower on Aeroelastic Loads of the Wind Turbine

**Junbo Hao \*, Zedong Wang, Wenwu Yi, Yan Chen \* and Jiyao Chen**

Institute of Energy and Environmental Science, Shantou University, Shantou 515063, China;
19zdwang@stu.edu.cn (Z.W.); 19wwyi@stu.edu.cn (W.Y.); 20symeng1@stu.edu.cn (J.C.)

\* Correspondence: 19jbhao@stu.edu.cn (J.H.); ychen@stu.edu.cn (Y.C.)

**Abstract:** The finite element discretization of a tower system based on the two-node Euler-Bernoulli beam is carried out by taking the cubic Hermite polynomial as the form function of the beam unit, calculating the structural characteristic matrix of the tower system, and establishing the wind turbine-nacelle-tower multi-degree-of-freedom finite element numerical model. The equation for calculating the aerodynamic load for any nacelle attitude angle is derived. The effect of the flexible tower vibration feedback on the aerodynamic load of the wind turbine is studied. The results show that, when the stiffness of the tower is large, the effect of having tower vibration feedback or not on the aeroelastic load of the wind turbine is small. For the more flexible tower system, wind-induced vibration time-varying feedback will cause larger aeroelastic load variations, especially the top of the tower overturning moment, thus causing a larger impact on the dynamic behavior of the tower downwind and crosswind. As the flexibility of the tower system increases, the interaction between tower vibration and pneumatic load is also gradually increasing. Taking into account the influence of flexible towers on the aeroelastic load of a wind turbine can help predict the pneumatic load of a wind turbine more accurately and improve the efficiency of wind energy utilization on the one hand and analyze the dynamic behavior of the flexible structure of a wind turbine more accurately on the other hand, which is extremely beneficial to the structural optimization of wind turbine.

**Keywords:** flexible tower; nacelle attitude feedback (NAF); dynamic response; aerodynamic load

## 1. Introduction

As the supporting structure of the wind turbine, the safety and reliability of the tower are directly related to the operation state of the wind turbine [1]. During the operation of the wind turbine, the dynamic behavior of the blades and the tower influence each other. To obtain more accurate dynamic response results, the blade-tower coupling effect must be considered in the dynamic analysis of wind turbines [2,3]. For wind turbines with complex structures [4,5], to effectively analyze their dynamic behavior, it is necessary to establish a simplified model of their dynamics [6,7]. The accurate calculation of tower wind load has an important influence on the accurate assessment of tower dynamic response. The nacelle is one of the main components of a wind turbine, and different nacelle attitude angles correspond to different aerodynamic loads of the wind turbine.

In this research, a simplified multi-degree-of-freedom numerical model of the horizontal axis wind turbine tower system, including the wind turbine-nacelle-tower foundation [8], was established and used to study the dynamic behavior of offshore HAWT tower system [9]. At the same time, based on the blade element momentum theory, a calculation method for the aerodynamic load of rotating blades including time-varying nacelle attitude feedback is proposed. This method uses a set of Euler angles to describe the time-varying attitude of the horizontal-axis wind turbine nacelle. This paper takes four basic cases as the research object and calculates the load on the top of the tower under the action of the flexibility of the foundation. These loads are used to analyze the dynamic response of the



tower system and discuss the influence of the flexibility characteristics of the foundation on the dynamic characteristics of the tower.

## 2. Analytical Study

### 2.1. Equations of Motion for Vibrating Systems

To effectively analyze the inherent characteristics of the wind turbine tower system, a simplified model of the tower system was developed, as shown in Figure 1. The model simplifies the wind turbine nacelle and the wind wheel as concentrated masses $M_s$, which are attached to the top of the tower. Additionally, the flexibility of the HAWT foundation is simulated using rotating springs and horizontal tension springs.

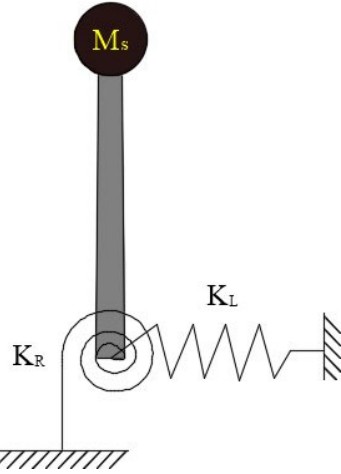

**Figure 1.** Simplified model of the tower system.

Based on the geometric properties of the structure, material properties, and external excitation properties, the multi-degree-of-freedom equation of motion is generally obtained as:

$$M\ddot{x}(t) + C\dot{x}(t) + Kx(t) = P(t) \tag{1}$$

where $M$ is the mass matrix; $C$ is the structural damping matrix, and $K$ is the stiffness matrix, which is the external load acting on the tower.

### 2.2. Tower Finite Element Modeling

In this paper, the top mass of the tower is simplified to a concentrated mass block and attached to the top of the tower. As for the thin-walled tower with a variable cross-section, a two-node Euler-Bernoulli beam is used to discretize the finite element of the wind turbine tower, and a cubic Hermite polynomial is chosen as the form function of each beam unit. Figure 2 shows a beam unit of the wind turbine tower with length $l$, mass per unit length $m(x)$, and bending stiffness $EI(x)$. The two nodes of this unit are located at the two endpoints of the beam; through these two nodes, the finite unit can be assembled into a single structure.

If only plane displacement is considered, each node of the beam unit has only two degrees of freedom, which are lateral displacement and rotation. The relationship between the displacement of the beam unit and the four degrees of freedom can be expressed as:

$$u(x,t) = \sum_{i=1}^{4} u_i(t)\psi_i(x) \tag{2}$$

where $\psi_i(x)$ is defined as the unit displacement due to the occurrence of unit displacement $u_i(x)$ while keeping the other degrees of freedom at zero. $\psi_i(x)$ satisfies the following boundary conditions:

$$\left.\begin{array}{l} i = 1 : \psi_1(0) = 1, \psi_1'(0) = \psi_1(l) = \psi_1'(l) = 0 \\ i = 2 : \psi_2'(0) = 1, \psi_2(0) = \psi_2(l) = \psi_2'(l) = 0 \\ i = 3 : \psi_3(l) = 1, \psi_3(0) = \psi_3'(0) = \psi_3'(l) = 0 \\ i = 4 : \psi_4'(0) = 1, \psi_4(0) = \psi_4(0) = \psi_4(l) = 0 \end{array}\right\} \tag{3}$$

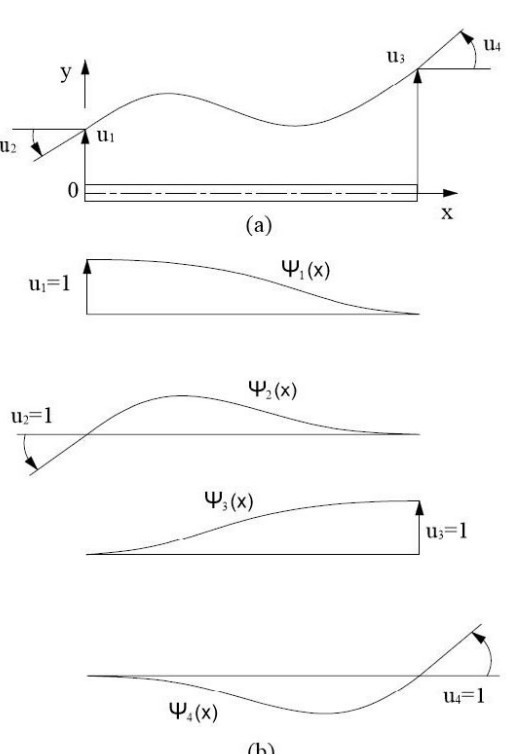

**Figure 2.** Beam cell model, (**a**) beam cell degrees of freedom, (**b**) interpolation function.

The equilibrium equation for a beam of equal cross-section considering only the load acting at both ends without taking into account the shear deformation is:

$$EI\frac{d^4u}{dx^4} = 0 \tag{4}$$

Its general solution is: $u(x) = a_1 + a_2\left(\frac{x}{l}\right) + a_3\left(\frac{x}{l}\right)^2 + a_4\left(\frac{x}{l}\right)^3$. The constant $a_i$ can be determined by substituting the boundary conditions, which gives:

$$\left.\begin{array}{l} \psi_1(x) = 1 - 3\left(\frac{x}{l}\right)^2 + 2\left(\frac{x}{l}\right)^3 \\ \psi_2(x) = l\left(\frac{x}{l}\right) - 2l\left(\frac{x}{l}\right)^2 + l\left(\frac{x}{l}\right)^3 \\ \psi_2(x) = 3\left(\frac{x}{l}\right)^2 - 2\left(\frac{x}{l}\right)^3 \\ \psi_4(x) = -l\left(\frac{x}{l}\right)^2 + l\left(\frac{x}{l}\right)^3 \end{array}\right\} \tag{5}$$

where $\psi_1(x)$, $\psi_2(x)$, $\psi_3(x)$, and $\psi_4(x)$ are the cubic Hermite interpolation functions, which ensure the continuity of deflection and cornering between beam units at the boundary.

Using the principle of imaginary displacement for a beam unit with bending stiffness $EI(x)$ and length $l$, the general expressions for the stiffness influence coefficient $k_{ij}$ and the mass influence coefficient $m_{ij}$ of the beam unit can be derived as follows:

$$k_{ij} = \int_0^l EI(x)\psi_i''(x)\psi_j''dx \tag{6}$$

$$m_{ij} = \int_0^l m(x)\psi_i(x)\psi_jdx \tag{7}$$

For cells with $EI(x) = EI$ and uniformly distributed masses ($m(x) \equiv \overline{m}$), when $i, j$ = 1, 2, 3, and 4, the analytical solutions of the cell stiffness matrix and the consistent mass matrix can be calculated as:

$$\overline{k_e} = \frac{EI}{l^3} \begin{bmatrix} 12 & 6l & -12 & 6l \\ 6l & 4l^2 & -6l & 2l^2 \\ -12 & -6l & 12 & -6l \\ 6l & 2l^2 & -6l & 4l^2 \end{bmatrix} \tag{8}$$

$$\overline{m_e} = \frac{ml}{420} \begin{bmatrix} 156 & 22l & 54 & -13l \\ 22l & 4l^2 & 13l & -3l^2 \\ 54 & 13l & 156 & -22l \\ -13l & -3l^2 & -22l & 4l^2 \end{bmatrix} \tag{9}$$

The wind turbine tower is always subjected to the gravitational force of the top mass of the tower, and this axial pressure also has a certain degree of influence on the stiffness of the tower. Using Hermite cubic polynomials as an interpolation function yields the general form of the unit geometric stiffness influence coefficient:

$$k_{Gij} = \int_0^l N(x)\psi_i'(x)\psi_j'dx \tag{10}$$

where $N(x)$ denotes the axial force acting on the unit. When $N(x) = N_G$ is constant, the consistent geometric stiffness matrix of the unit is obtained as:

$$\overline{k_{Ge}} = \frac{N_G}{30l} \begin{bmatrix} 36 & 3l & -36 & 3l \\ 3l & 4l^2 & -3l & -l^2 \\ -36 & -3l & 36 & -3l \\ 3l & -l^2 & -3l & 4l^2 \end{bmatrix} \tag{11}$$

The axial force along the full length of the unit is assumed to be constant. By superimposing the above two effects, elastic and geometric, the total stiffness matrix of the structural unit is obtained, expressed as:

$$\overline{k} = \overline{k_e} - \overline{k_{Ge}} \tag{12}$$

where the negative sign indicates that the presence of axial load $N(x)$ increases the deflection of the tower.

If units $m$, $n$, and $p$ are connected to node $i$ of the structure, the stiffness coefficient of this node can be calculated using the direct stiffness method as:

$$\hat{\overline{k}}_{ii} = \hat{\overline{k}}_{ii}^{(m)} + \hat{\overline{k}}_{ii}^{(n)} + \hat{\overline{k}}_{ii}^{(p)} \tag{13}$$

where the notation '^' indicates the coefficient in the overall coordinate system. The stiffness matrix of the whole structure can be obtained by Equation (13). Similarly, the consistent mass matrix of the whole structure can be calculated.

The damping characteristics of structures generally need to be determined by experimental methods, but it would be quite difficult and impractical to obtain the damping of each structure by experimental tests. Therefore, the dynamic properties of a structure are often analyzed by using a classical damping model. In this paper, the Rayleigh damping model is used, which integrates the effects of mass-proportional damping and stiffness-proportional damping, and its calculation equation is as follows:

$$C = \eta_0 M + \eta_1 K \tag{14}$$

where $\eta_0$ and $\eta_1$ are scaling factors.

$$\begin{pmatrix} \eta_0 \\ \eta_1 \end{pmatrix} = \frac{2\omega_m\omega_n}{\omega_n^2 - \omega_m^2} \begin{pmatrix} \omega_n & -\omega_m \\ -1/\omega_n & 1/\omega_m \end{pmatrix} \begin{pmatrix} \xi_m \\ \xi_n \end{pmatrix} \tag{15}$$

where $\omega_m$ and $\omega_n$ are the m-order and n-order frequencies of the vibrating system, and $\xi_m$ and $\xi_n$ are the damping ratios of the corresponding modal frequencies [8].

When $\xi_m = \xi_n = \xi$, it can be simplified as follows:

$$\begin{pmatrix} \eta_0 \\ \eta_1 \end{pmatrix} = \frac{2\xi}{\omega_m + \omega_n} \begin{pmatrix} \omega_m\omega_n \\ 1 \end{pmatrix} \tag{16}$$

In practical engineering applications, $\omega_m$ is usually taken as the fundamental frequency of the vibrating system, and $\omega_n$ is taken as the higher order frequency that has a significant effect on the dynamic behavior of the system. When the modal damping and modal frequency of the tower system are known, its damping matrix can be determined by Equations (15) and (16). After calculating the mass matrix, stiffness matrix and damping matrix of the whole tower, the multi-degree-of-freedom finite element numerical model of the tower system can be established by treating the top mass of the tower as a boundary condition [10].

### 2.3. Eigenvalue Analysis

The eigenvalue problem of the structure is to analyze its free vibration. To calculate the natural frequency and modal vibration pattern of the tower system, the effect of damping is generally neglected. Removing the damping matrix and external excitation of Equation (1), the free vibration equation of the tower can be obtained as follows:

$$M\ddot{x}(t) + Kx(t) = 0 \tag{17}$$

The characteristic equation of the tower can be solved as:

$$\left(K - \omega^2 M\right)\phi = 0 \tag{18}$$

where $\omega$ denotes the frequency of free vibration, and $\phi$ denotes the time-independent nth-order vector. $\phi = \begin{bmatrix} \phi_1 & \phi_2 & \cdots & \phi_n \end{bmatrix}$, $\phi_n$ denote the nth-order vibration column vector of the structure.

Wind turbines operating in different seas have different foundation characteristics. Adhikari et al. [11] studied and obtained classical values of foundation stiffness for three actual operating wind turbines. Based on this study, 12 foundation cases are selected in this paper to study the effect of flexible foundations on the towers' inherent frequency. The wind turbine numerical model parameters used for simulation in this paper were selected from the NREL 5 MW offshore horizontal axis wind turbine [12]. The first three orders of tower inherent frequencies are calculated as shown in Table 1.

From Table 1, it can be concluded that the rotational stiffness has a greater effect on the first two orders of the towers' natural frequency and a smaller effect on the third order of the towers' natural frequency. The horizontal stiffness of a tower on the natural frequency of the tower is opposite to the effect of rotational stiffness. For a tower with a rigid foundation, the first three orders of inherent frequency are also calculated, as shown in Table 2.

From Tables 1 and 2, it can be concluded that the flexible foundation of a tower has an important effect on the free vibration of the tower. Therefore, the flexibility of the tower needs to be taken into account in the dynamic analysis of offshore wind turbines.

**Table 1.** The first three orders of the towers' inherent frequency (unit: Hz).

| $K_R$ (GNm/rad) | $K_L$ (GN/m) | 1st | 2nd | 3rd |
|---|---|---|---|---|
| **10** | 0.5 | 0.20862 | 2.0824 | 6.8703 |
| 10 | 1 | 0.20874 | 2.0991 | 7.1053 |
| 10 | 5 | 0.20882 | 2.1124 | 7.2858 |
| 20 | 0.5 | 0.25229 | 2.2495 | 6.9903 |
| 20 | 1 | 0.25245 | 2.2725 | 7.2701 |
| 20 | 5 | 0.25249 | 2.2909 | 7.4874 |
| 50 | 0.5 | 0.29507 | 2.5064 | 7.2177 |
| 50 | 1 | 0.29533 | 2.5409 | 7.5925 |
| 50 | 5 | 0.29554 | 2.5683 | 7.8877 |
| 100 | 0.5 | 0.31463 | 2.6753 | 7.4038 |
| 100 | 1 | 0.31494 | 2.7180 | 7.8656 |
| 100 | 5 | 0.31519 | 2.7519 | 8.2307 |

**Table 2.** First three orders of inherent frequency of towers under a rigid foundation (unit: Hz).

| Modal | 1st | 2nd | 3rd |
|---|---|---|---|
| Natural frequency | 0.3391 | 3.0634 | 9.0983 |

### 2.4. Solving the Nacelle Attitude Angle Using Euler Angles

Attitude solutions are relatively common in computer graphics and aerospace, and Euler angles [13] have been widely used in the field of attitude description due to their simplicity and ease of use. A set of Euler angles, which are the yaw, pitch, and roll angles obtained from the rotation of the nacelle about the X, Y, and Z coordinate axes, are used to describe the attitude of the HAWT nacelle, as shown in Figure 3.

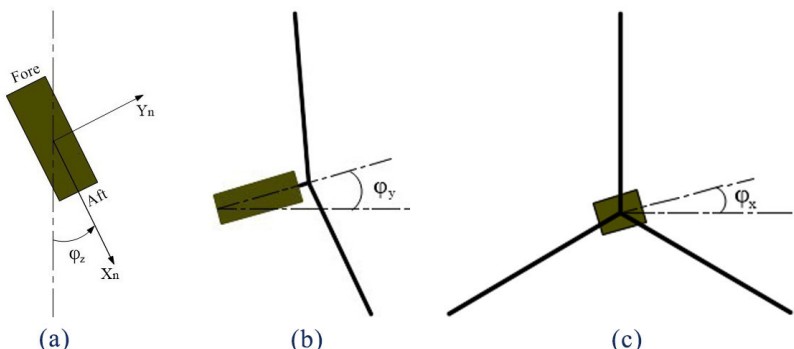

(a)          (b)          (c)

**Figure 3.** Nacelle attitude angle, (**a**) yaw, (**b**) pitch, (**c**) roll.

Assume that the nacelle transitions from one attitude to another in the sequence of transverse roll-pitch-yaw, wherein the transition matrix around the axis is:

$$R_1(\varphi_x) = \begin{bmatrix} 1 & 0 & 0 \\ 0 & \cos\varphi_x & -\sin\varphi_x \\ 0 & \sin\varphi_x & \cos\varphi_x \end{bmatrix} \tag{19}$$

$$R_2(\varphi_y) = \begin{bmatrix} \cos\varphi_y & 0 & \sin\varphi_y \\ 0 & 1 & 0 \\ -\sin\varphi_y & 0 & \cos\varphi_y \end{bmatrix} \tag{20}$$

$$R_3(\varphi_z) = \begin{bmatrix} \cos\varphi_z & -\sin\varphi_z & 0 \\ \sin\varphi_z & \cos\varphi_z & 0 \\ 0 & 0 & 1 \end{bmatrix} \tag{21}$$

The attitude matrix of the nacelle can be obtained as follows:

$$R_A = R_1(\varphi_x)R_2(\varphi_y)R_3(\varphi_z) \tag{22}$$

## 3. Calculation Method

### 3.1. Calculation of Aerodynamic Loads under Consideration of NAF

For any nacelle attitude angle, the horizontal axis wind turbine coordinate system shown in Figure 4 is established to facilitate the calculation of the spatial position and wind speed components of each wing section of the blade [14].

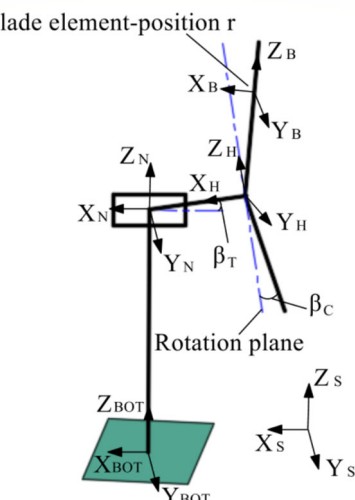

**Figure 4.** Horizontal axis wind turbine coordinate system.

1.  The inertial coordinate system $S$ coincides with the tower bottom coordinate system $S_{BOT}$, and the nacelle coordinate system $S_N$ coincides with the tower top coordinate system $S_{top}$.
2.  When the azimuth angle $\theta = 0$, the wind turbine rotation plane coordinate system $S_R$ coincides with the hub coordinate system $L_H$, and $S_R$ rotates with the azimuth angle $\theta$.
3.  The tower height is $L$, and the horizontal distance from the top of the tower to the hub is $L_H$.
4.  The spindle inclination angle and blade taper angle are $\beta_T$ and $\beta_C$.

The transformation relationship between the blade coordinate system and the inertial coordinate system exists both in terms of rotation and translation. The transformation between the coordinate systems is described using the chi-square coordinates [15] as $S = \begin{bmatrix} X & Y & Z & 1 \end{bmatrix}^T$. The transformation relationship between the coordinate systems is as follows:

$$S_N = K_L S \tag{23}$$

$$S_H = K_{LH}K_{\beta T}K_{DN}R_N S_N \tag{24}$$

$$S_R = K_\theta S_H \tag{25}$$

$$S_B = K_{\beta C}S_R \tag{26}$$

where: $K_L$ is the transformation matrix from the inertial coordinate system to the nacelle coordinate system; $K_{Lh}$ and sub $K_{\beta T}$ are the translation and rotation transformation matrices from the nacelle to the hub coordinate system, respectively; $R_N$ and $K_{DN}$ are the attitude matrix and deflection matrix of the nacelle, respectively; $K_\theta$ is the transformation

matrix from the hub to the rotating coordinate system of the wind wheel, and $K_{\beta C}$ is the transformation matrix from the wind wheel to the rotating blade.

$$K_L = \begin{bmatrix} 1 & 0 & 0 & 0 \\ 0 & 1 & 0 & 0 \\ 0 & 0 & 1 & -L \\ 0 & 0 & 0 & 1 \end{bmatrix} \tag{27}$$

$$K_{LH} = \begin{bmatrix} 1 & 0 & 0 & L_H/\cos\beta_T \\ 0 & 1 & 0 & 0 \\ 0 & 0 & 1 & 0 \\ 0 & 0 & 0 & 1 \end{bmatrix} \tag{28}$$

$$K_{\beta T} = \begin{bmatrix} \cos\beta_T & 0 & \sin\beta_T & 0 \\ 0 & 1 & 0 & 0 \\ -\sin\beta_T & 0 & \cos\beta_T & 0 \\ 0 & 0 & 0 & 1 \end{bmatrix} \tag{29}$$

$$K_{DN} = \begin{bmatrix} 1 & 0 & 0 & x_N \\ 0 & 1 & 0 & y_N \\ 0 & 0 & 1 & z_N \\ 0 & 0 & 0 & 1 \end{bmatrix} \tag{30}$$

$$K_\theta = \begin{bmatrix} 1 & 0 & 0 & 0 \\ 0 & \cos\theta & -\sin\theta & 0 \\ 0 & -\sin\theta & \cos\theta & 0 \\ 0 & 0 & 0 & 1 \end{bmatrix} \tag{31}$$

$$K_{\beta C} = \begin{bmatrix} \cos\beta_c & 0 & -\sin\beta_c & 0 \\ 0 & 1 & 0 & 0 \\ \sin\beta_c & 0 & \cos\beta_c & 0 \\ 0 & 0 & 0 & 1 \end{bmatrix} \tag{32}$$

From Equations (23) to (32), the transformation relation between the blade coordinate system and the inertial coordinate system can be obtained as:

$$S = K_L^{-1} R_N^{-1} K_{DN}^{-1} K_{\beta T}^{-1} K_{LH}^{-1} K_\theta^{-1} K_{\beta C}^{-1} S_B \tag{33}$$

Therefore, the coordinate component $\begin{bmatrix} X & Y & Z \end{bmatrix}^T$ in the inertial coordinate system for the leaf element coordinates $\begin{bmatrix} X_B & Y_B & Z_B \end{bmatrix}^T = \begin{bmatrix} 0 & 0 & r \end{bmatrix}^T$ is calculated by Equation (34).

$$\begin{bmatrix} X \\ Y \\ Z \\ 1 \end{bmatrix} = K_L^{-1} R_N^{-1} K_{DN}^{-1} K_{\beta T}^{-1} K_{LH}^{-1} K_\theta^{-1} K_{\beta C}^{-1} \begin{bmatrix} 0 \\ 0 \\ r \\ 1 \end{bmatrix} \tag{34}$$

The coordinate component of the wind speed $U_S = \begin{bmatrix} U & 0 & 0 \end{bmatrix}^T$ in the inertial coordinate system in the blade coordinate system can be calculated by the following equation.

$$\begin{bmatrix} U_{BX} \\ U_{BY} \\ U_{BZ} \\ 1 \end{bmatrix} = R_N K_{\beta T} K_\theta K_{\beta C} \begin{bmatrix} U \\ 0 \\ 0 \\ 1 \end{bmatrix} \tag{35}$$

The load on the wind turbine tower is mainly from the inertial load on the top mass of the tower and the aerodynamic load on the rotating blades. The blade load and tower

top load can be decomposed into component forces and bending moments along the three coordinate axes, as shown in Figure 5.

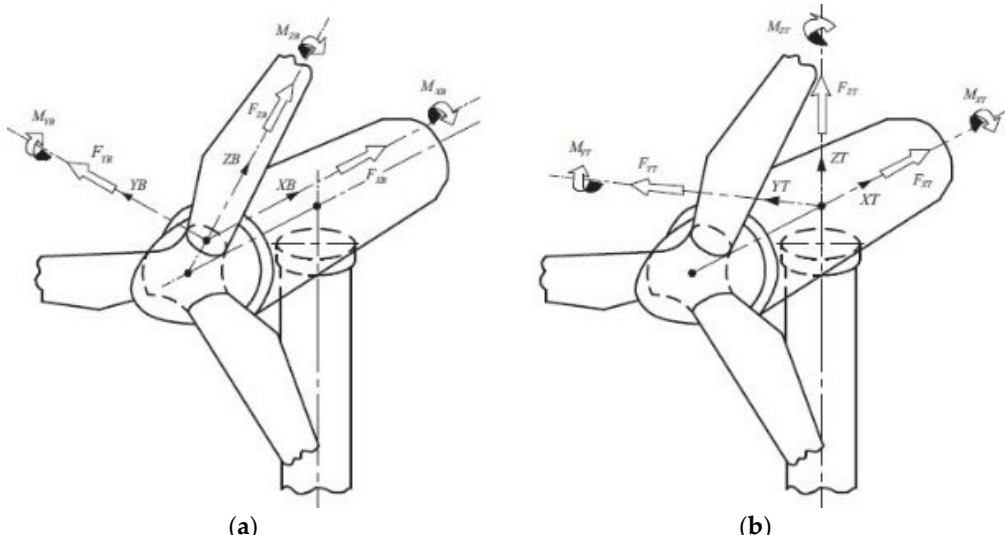

(**a**)                                        (**b**)

**Figure 5.** Wind turbine load coordinate system [16]. (**a**) Blade coordinate system. (**b**) Tower coordinate system.

When the wind component of turbulent wind in the lobe element coordinate system is obtained, the angle of entry and relative wind speed of the incoming wind on the lobe element can be calculated by the following equation.

$$\phi = tg\{U_{BX}(1-a)/(U_{BY} + wr(1+a')\cos\beta_C)\} \tag{36}$$

$$W = \sqrt{\left(U_{BX}(1-a)^2 + U_{BY} + wr(1-a')\cos\beta_C\right)^2} \tag{37}$$

In this paper, the dynamic response of the tower in the forward–backward direction and the left–right direction is investigated. The aerodynamic forces and aerodynamic moments on the blades were calculated using BEM theory [16] with the following equations.

$$dF_{XB} = \frac{1}{2}\rho W^2 c(C_L \sin\phi - C_D \cos\phi)dr \tag{38}$$

$$dF_{YB} = \frac{1}{2}\rho W^2 c(C_L \cos\phi + C_D \sin\phi)dr \tag{39}$$

$$dM_{XB} = \frac{1}{2}\rho W^2 c(C_L \sin\phi - C_D \cos\phi)rdr \tag{40}$$

$$dM_{YB} = \frac{1}{2}\rho W^2 c(C_L \sin\phi - C_D \cos\phi)rdr \tag{41}$$

where $C_L$ and $C_D$ denote the lift and drag coefficients of the cross-sectional airfoil; $\rho$ is the air density, and $c$ is the chord length of the blade cross-sectional airfoil.

*3.2. Tower Load Calculation*

The aerodynamic forces $F$ and aerodynamic moments $M$ on the wind turbine are transformed by coordinates to obtain the aerodynamic loads acting on the tower.

$$\begin{bmatrix} F_{BXT} & F_{BYT} & F_{BZT} & 1 \end{bmatrix}^T = C_{BT}\begin{bmatrix} F_B & 1 \end{bmatrix}^T \tag{42}$$

$$\begin{bmatrix} M_{BXT} & M_{BYT} & M_{BZT} & 1 \end{bmatrix}^T = C_{BT}\begin{bmatrix} M_B & 1 \end{bmatrix}^T \tag{43}$$

where $C_{BT} = R_N^{-1}K_{\beta T}^{-1}K_\theta^{-1}K_{\beta C}^{-1}$ is the transformation matrix.

The eccentricity of the top mass center of the wind turbine tower with the tower center will produce the overturning moment and pitching moment acting on the tower. In the tower top coordinate system, assuming the coordinates of the tower top mass center as $\begin{pmatrix} X_{TE} & Y_{TE} & Z_{TE} \end{pmatrix}$, the tower top bending moment caused by the eccentricity of the tower top mass can be calculated by the following equation.

$$M_{YE} = M_s g X_{TE} \tag{44}$$

$$M_{XE} = M_s g Y_{TE} \tag{45}$$

where $M_s$ is the tower top mass, and $g$ is the acceleration of gravity.

The final total load of the tower can be obtained by combining the wind turbine pneumatic load and the eccentric moment of the top mass of the tower on the tower.

### 3.3. Solving Structural Equations of Motion

For a general external excitation $P(t)$, the Newmark method is used in this paper to solve the multi-degree-of-freedom equations of motion.

The Newmark method approximates the velocity and displacement of the system at the moment $(t + \Delta t)$ by two assumptions.

$$\dot{x}_{t+\Delta t} = \dot{x}_t + \left[ (1 - \beta)\ddot{x}_t + \beta\ddot{x}_{t+\Delta t} \right] \Delta t \tag{46}$$

$$x_{t+\Delta t} = x_t + \dot{x}_t \Delta t + \left[ \left( \frac{1}{2} - \alpha \right) \ddot{x}_t + \alpha\ddot{x}_{t+\Delta t} \right] \Delta t^2 \tag{47}$$

where $\alpha$ and $\beta$ are the parameters adjusted according to the accuracy and stability requirements of the integration. When $\alpha = 1/6$, $\beta = 1/2$, it is the linear acceleration method; when $\alpha = 1/4$, $\beta = 1/2$, it is the average acceleration method; when $\alpha = 1/2$, $\beta = 1/2$, it is the central difference method; when $\alpha = 1/8$, $\beta = 1/2$, it is the variable acceleration method.

From Equations (44) and (45), we can solve that:

$$\ddot{x}_{t+\Delta t} = \frac{1}{\alpha\Delta t^2}(x_{t+\Delta t} - x_t) - \frac{1}{\alpha\Delta t}\dot{x}_t - \left( \frac{1}{2\alpha} - 1 \right)\ddot{x}_t \tag{48}$$

$$\dot{x}_{t+\Delta t} = \dot{x}_t + (1 - \beta)\Delta t\ddot{x}_t + \beta\Delta t\ddot{x}_{t+\Delta t} \tag{49}$$

When $\beta \geq 0.5$ and $\alpha \geq 0.25(0.5 + \beta)^2$, the Newmark method is the unconditionally stable format. The equations of motion can be solved by the following steps.

1. Initial Calculation

    1. Firstly, the overall characteristic matrices *K*, *M* and *C* of the vibrating system are calculated.
    2. From the initial conditions $x_0$ and $\dot{x}_0$, $\ddot{x}_0$ is calculated.

$$\ddot{x}_0 = M^{-1}\left( F_0 - C\dot{x}_0 - Kx_0 \right) \tag{50}$$

    3. Step $\Delta t$ is selected with parameters $\alpha$, $\beta$ and calculate the integration constants.

$$A_1 = \frac{1}{\alpha\Delta t^2}, A_2 = \frac{\beta}{\alpha\Delta t}, A_3 = \frac{1}{\alpha\Delta t}, A_4 = \left( \frac{1}{2\alpha} - 1 \right), A_5 = \frac{\Delta t}{2}\left( \frac{\beta}{\alpha} - 2 \right), A_6 = \left( \frac{\beta}{\alpha} - 1 \right)$$

    4. The equivalent stiffness matrix $\overline{K}$ is calculated.

$$\overline{K} = K + A_1 M + A_2 C \tag{51}$$

2. For each time step

1.　The equivalent load vector $\overline{F}$ at moment $t + \Delta t$ is determined.

$$F = F_{t+\Delta t} + \left[A_1 x_t + A_3 \dot{x}_t + A_4 \ddot{x}_t\right]M + \left[A_2 x_t + A_6 \dot{x}_t + A_3 \ddot{x}_t\right]C \tag{52}$$

2.　The displacement at moment $t + \Delta t$ is calculated.

$$x_{t+\Delta t} = \overline{K}^{-1}\overline{F}_{t+\Delta t} \tag{53}$$

3.　The velocity at time $t + \Delta t$ and the acceleration are solved.

$$\ddot{x}_{t+\Delta t} = \frac{1}{\alpha \Delta t^2}(x_{t+\Delta t} - x_t) - \frac{1}{\alpha \Delta t}\dot{x}_t - \left(\frac{1}{2\alpha} - 1\right)\ddot{x}_t \tag{54}$$

$$\dot{x}_{t+\Delta t} = \dot{x}_t + (1 - \beta)\Delta t \ddot{x}_t + \beta \Delta t \ddot{x}_{t+\Delta t} \tag{55}$$

### 3.4. Dynamic Response Analysis Process

The dynamic response of the tower system is calculate as shown in Figure 6, taking into account the effects of nacelle attitude feedback and foundation flexibility of the foundation.

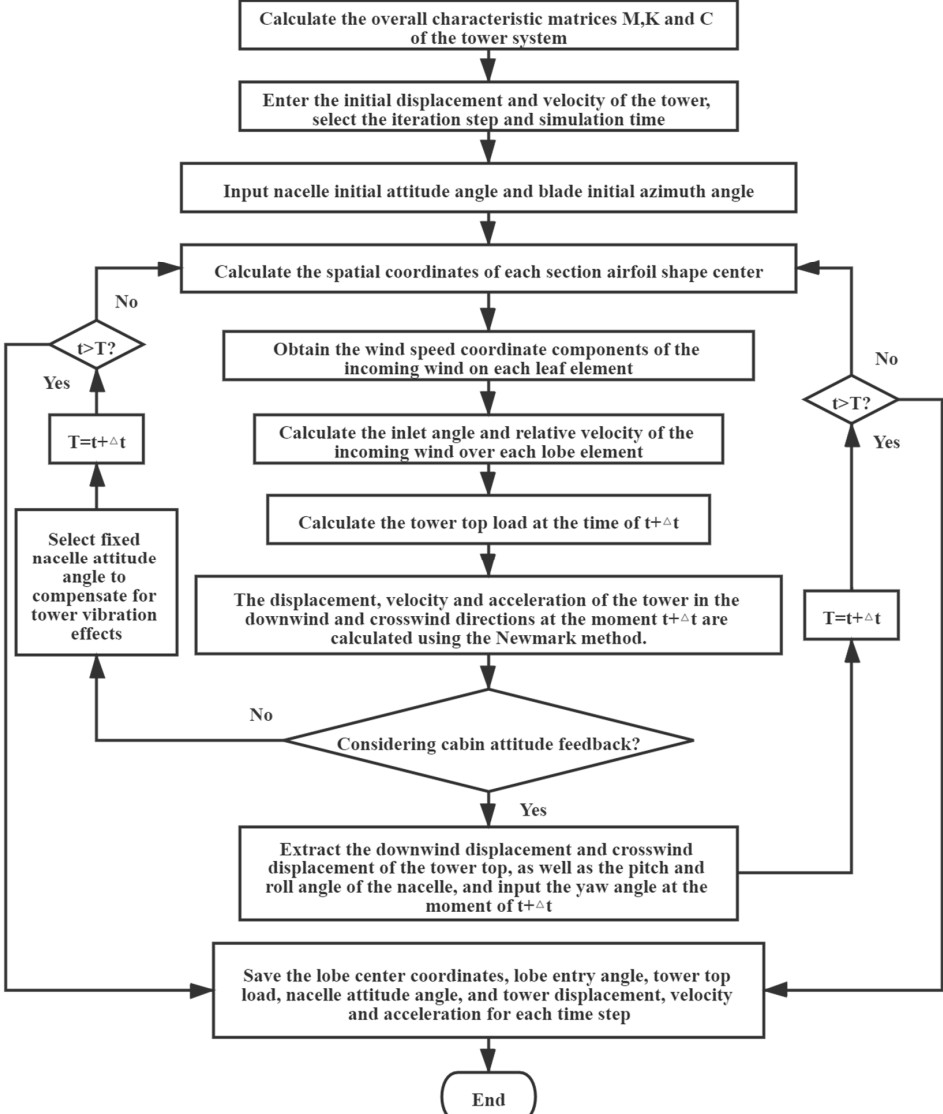

**Figure 6.** Dynamic response solving process of flexible tower.

## 4. Flexible Tower Vibration Feedback Analysis

### 4.1. Tower Load Error Analysis

Under the action of wind load, the tower will vibrate, which will cause the pitch and roll angle of the nacelle to change, and the change of attitude angle will in turn affect the load calculation of the rotating blade, which is a cyclic process. In this paper, we design a method to analyze the dynamics under the action of dynamic structural parameter feedback, and use MATLAB to prepare the corresponding program to explore the laws through numerical examples [9]. In analyzing the effect of flexible pylons on aeroelastic loads, the following assumptions are made first.

1.  The nacelle and the tower have only yaw motion.
2.  The center of mass of the nacelle coincides with the center of the tower top.

The deflection of the tower top in downwind direction and crosswind direction will be equal to the fore-and-aft displacement and left–right displacement of the nacelle respectively, and the angle of rotation of the tower top in fore-and-aft direction and left–right direction will be equal to the pitch angle and cross-roll angle of the nacelle, respectively. The relationship between the tower top turning angle and the nacelle attitude angle is shown in Figure 7.

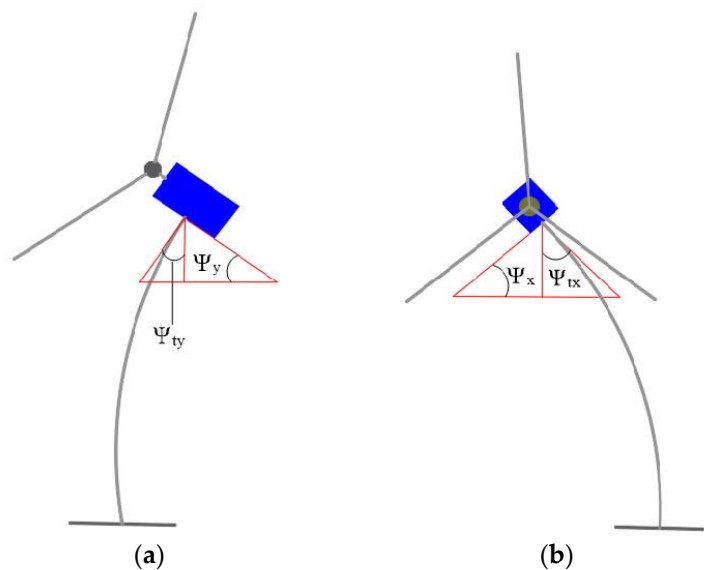

(**a**)                (**b**)

**Figure 7.** Relationship between tower turning angle and cabin attitude. (**a**) Forward and backward. (**b**) Left and right.

$\psi_{ty}$ and $\psi_{tx}$ denote the angle of rotation of the top of the tower in the front-to-back direction and the left-to-right direction, respectively. Based on the assumptions of this paper, there are $\psi_y = \psi_{ty}$, $\psi_x = \psi_{tx}$ holds at the same time. The load calculation formula for any nacelle attitude angle derived in this paper is applied to the study of the effect of the flexible tower on the aeroelastic load of the wind turbine. In the steady-state yaw condition, when considering the effect of the flexible tower vibration feedback on the wind turbine aeroelastic load, it is necessary to calculate the large feedback effect of the tower top front-to-back and left-right directional turning angle and displacement at the same time, and it is also necessary to consider the two-way feedback effect of the nacelle's cross-roll angle, pitch angle, downwind tower top displacement, and cross-wind tower top displacement at the same time, and its calculation process is shown in Figure 6. At the same time, the calculation flow without considering the feedback effect is also incorporated into this procedure, only when the feedback is not considered, the influence of tower vibration on the structural parameters needs to be compensated.

To simulate the roll of tower vibration feedback in the dynamic response of flexible towers, simulation analysis is performed for towers with different flexibility. First, the other parameters of the towers are maintained consistent with the NREL 5 MW wind turbine; the geometric parameters of the wind turbine tower and the basic operating parameters of the wind turbine are shown in Table 3. The increase of its flexibility is simulated by increasing the height of the tower, and the parameters of the tower used for numerical simulation are shown in Table 4.

**Table 3.** 5 MW wind turbine parameters.

| Name | Value | Name | Value |
|---|---|---|---|
| Rated power | 5000 KW | Cone angle | 2.5° |
| Rated wind speed | 11.4 m/s | Rated rotor speed | 12.1 rpm |
| Cut-in wind speed | 3 m/s | Rotor diameter | 126 m |
| Cut-out wind speed | 25 m/s | Tower mass density | 8500 kg/m$^3$ |
| Blade length | 61.5 m | Tower's modulus of elasticity | $2.1 \times 10^{11}$ N/m$^2$ |
| Blade mass | 17,740 kg | Structural-damping ratio | 1% |
| Blade number | 3 | Tower height | 86.7 m |
| Hub height | 90 m | Tower-base diameter | 6 m |
| Hub mass | 56,780 kg | Tower-base thickness | 0.0351 m |
| Nacelle mass | 240,000 kg | Tower-top diameter | 3.87 m |
| Shaft tilt | 5° | Tower-top thickness | 0.0247 |

**Table 4.** Ultra-high flexible tower parameters.

| Height (m) | Modulus of Elasticity (N/m$^2$) | Diameter and Wall Thickness of Tower Top (m) | Diameter and Wall Thickness of Tower Bottom (m) | Structural Damping Ratio |
|---|---|---|---|---|
| 86.7 | $2.1 \times 10^{11}$ | 3.87, 0.247 | 6, 0.351 | 1% |
| 200 | $2.1 \times 10^{11}$ | 3.87, 0.247 | 6, 0.351 | 1% |
| 300 | $2.1 \times 10^{11}$ | 3.87, 0.247 | 6, 0.351 | 1% |

The relative errors of the tower top load are calculated separately for the two cases of considering feedback and not considering feedback; the comparative analysis results of longitudinal thrust and transverse thrust are shown in Figure 8.

As can be seen from Figure 8, when the tower stiffness is relatively large, the effect on the wind turbine pneumatic load is small when considering the tower bending-bending coupling vibration feedback. When the tower flexibility increases, its load relative error increases rapidly, especially the transverse thrust; for each flexible tower, its relative error amplitude is almost maintained at about two times the relative error of the longitudinal thrust, which will have a significant increase in impact for the more flexible tower, and cause a large impact on the power of the wind turbine.

To investigate the law of mutual influence between tower vibration and pneumatic load, we compared the relative errors of tower top angle and displacement under three tower heights, and the simulation results are shown in Figures 9 and 10.

From Figures 9 and 10, it can be seen that, when the flexibility of the support structure is large, the maximum errors of displacement and the turning angle of the tower top downwind are both around 8%, while the maximum deviations of the crosswind direction are both over 12%. It can be seen that, when the flexibility of the support structure is large, ignoring the effect of the time-varying nacelle attitude feedback will make the analysis of the dynamic less accurate. It can also be seen that the relative errors of displacement and the turning angle at the top of the tower remain the same in terms of trend and magnitude. It can be seen that when the other structural parameters (blade, nacelle mass, etc.) are constant, the degree of influence of the tower bending-bending coupling bidirectional feedback on the dynamical behavior of the tower is positively correlated with the flexibility of the support structure.

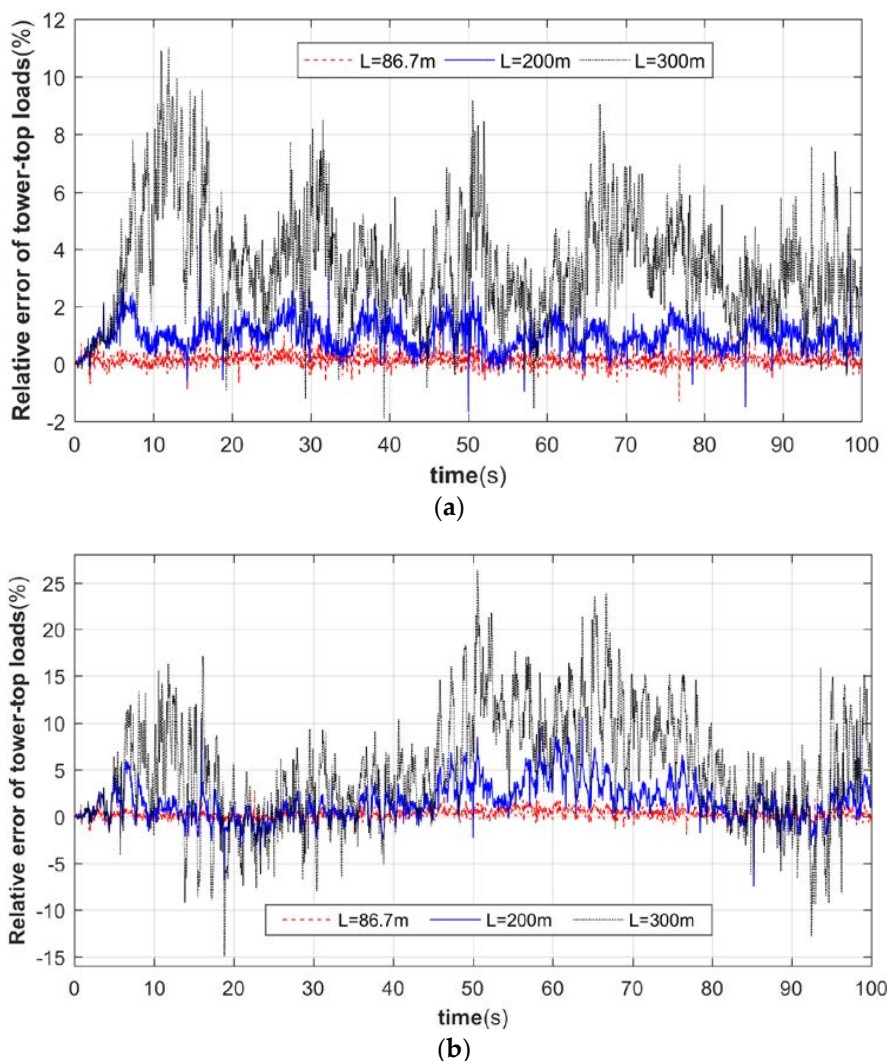

**Figure 8.** Relative error analysis of tower top load. (**a**) Longitudinal thrust. (**b**) Lateral thrust.

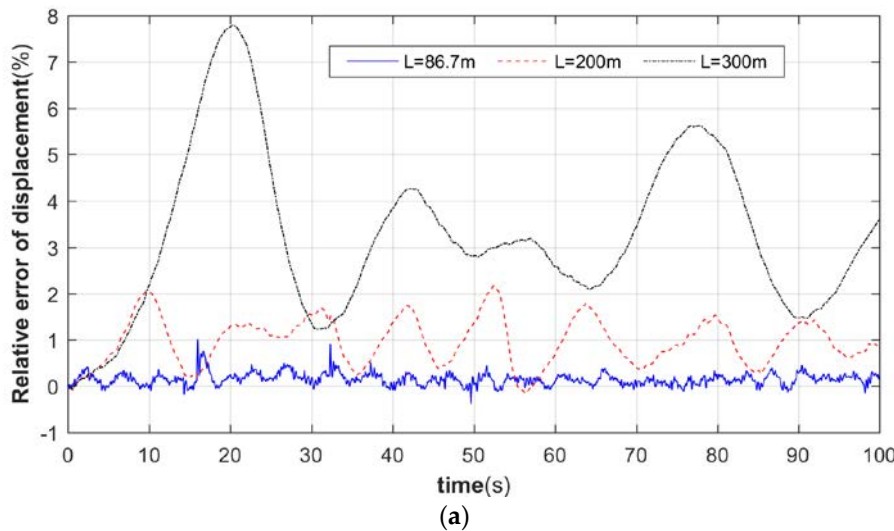

**Figure 9.** *Cont.*

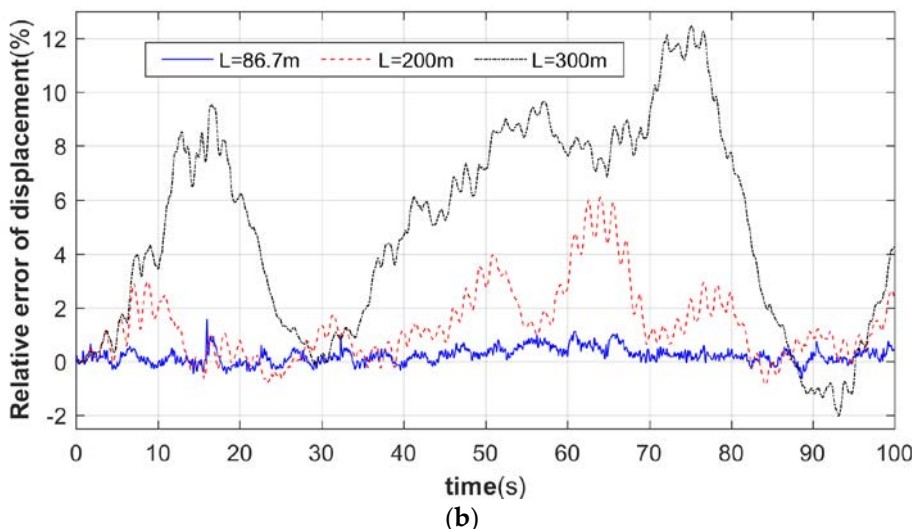

**Figure 9.** Relative error of tower top angle. (**a**) Pitch angle. (**b**) Lateral camber angle.

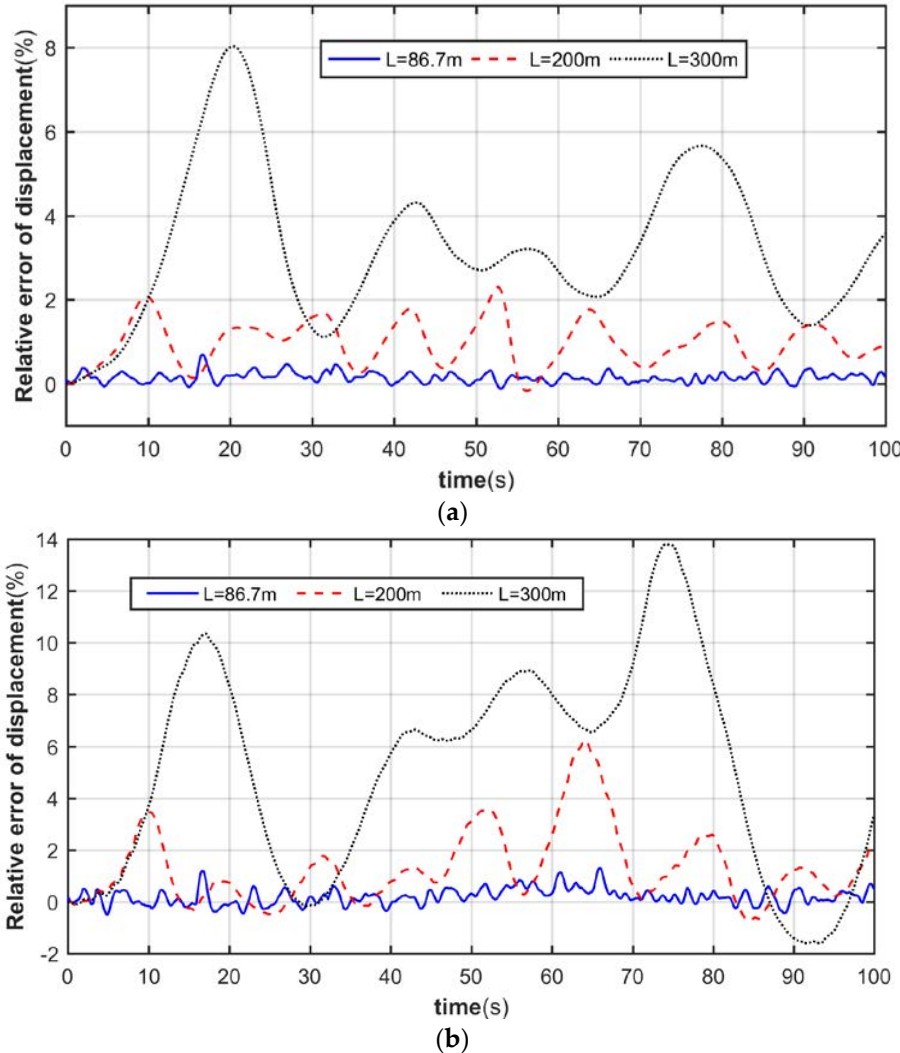

**Figure 10.** Relative error of tower top displacement. (**a**) Longitudinal direction. (**b**) Crosswind direction.

*4.2. Dynamic Response Analysis of Tower System*

Four foundation cases were selected to study the effect of flexible foundations on tower loads. The wind condition used is a steady state wind with 11.4 m/s and a turbulence intensity of 16.4946%.

For each of the four foundation cases in Table 5, the wind load response of the tower system is solved, and the effect of the foundation rotational stiffness on the tower vibration is explored. When considering the flexibility, the wind turbine tower top response time equations are shown in Figures 11 and 12.

**Table 5.** Simulation analysis foundation case.

|  | $K_L$ (GN/m) | $K_R$ (GNm/rad) |
|---|---|---|
| Case 1 | 1 | 10 |
| Case 2 | 1 | 20 |
| Case 3 | 1 | 50 |
| Case 4 | Rigid foundation | |

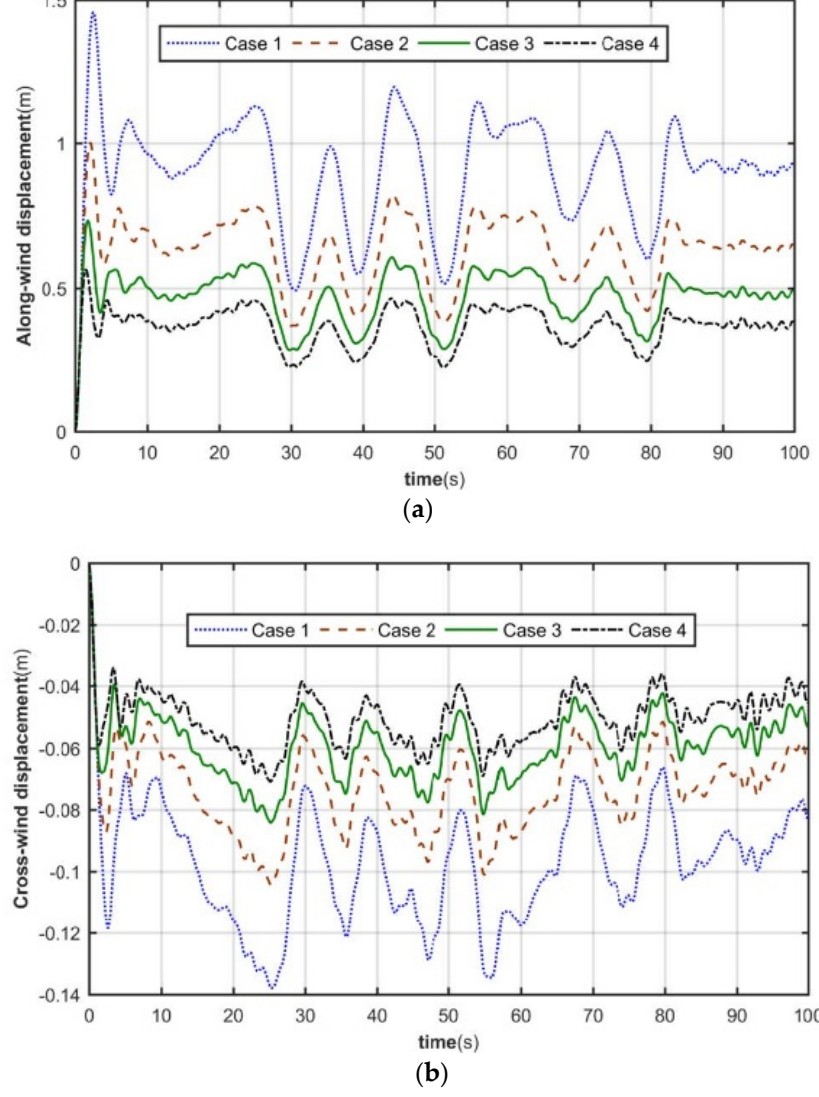

**Figure 11.** Time course of tower top displacement. (**a**) Downwind. (**b**) Sidewind direction.

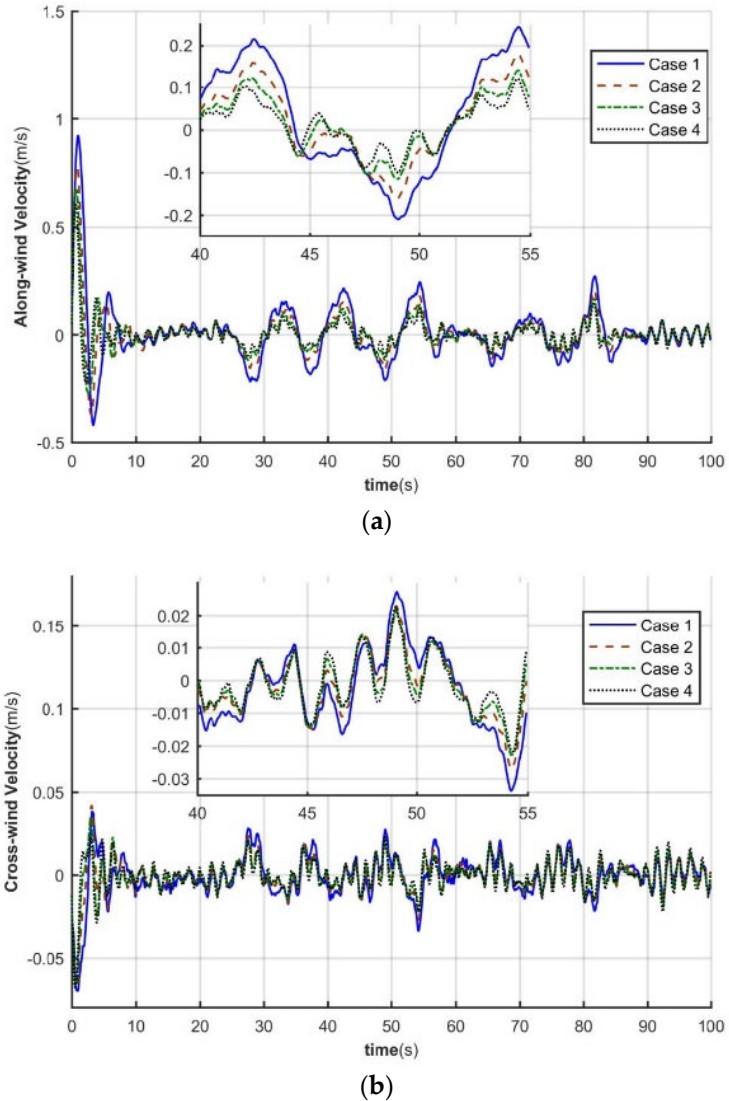

**Figure 12.** Time course of tower top speed (**a**) Downwind (**b**) Sidewind direction.

As shown in Figure 11, the maximum displacements of the tower top in the downwind direction are 1.2 m, 0.82 m, 0.61 m, and 0.46 m. The maximum lateral displacements of the tower top are 0.14 m, 0.1 m, 0.08 m, and 0.07 m. It can be seen that the deflection of the tower top increases greatly when considering flexibility. Compared with the rigid foundation, the maximum tower top displacement of Case 1, Case 2, and Case 3 increases nearly 161%, 78%, and 32% in the downwind direction and nearly 100%, 42%, and 14% in the sidewind direction, respectively.

As shown in Figure 12, the tower top velocity response is greatly increased when considering flexibility. Compared with the rigid foundation, the tower top velocities of Case 1, Case 2, and Case 3 increase by nearly 88%, 41%, and 14% in the downwind direction, and by nearly 55%, 23%, and 4% in the sidewind direction, respectively.

From Figures 11 and 12, it can be seen that the tower top displacement and velocity response amplitude increase with decreasing base stiffness when considering base stiffness. Moreover, when the rotational stiffness gradually increases, the closer to the rigid foundation, the smaller the decrease of the top displacement and velocity response amplitude.



## 5. Conclusions

(1) The finite element discretization of the pylon system is carried out with a two-node Euler-Bernoulli beam; the cubic Hermite polynomial is taken as the form function of the beam unit; the structural characteristic matrix of the pylon system is calculated, and its multi-degree-of-freedom finite element model is established. (2) The calculation formula of the effect of the nacelle attitude angle on the aerodynamic load of the blade is obtained. (3) When the tower stiffness is large, the effect of having or not having tower vibration feedback on the calculation of the wind turbine aeroelastic load is small. (4) For a more flexible tower system, wind-induced vibration time-varying feedback will cause a larger aeroelastic load variation, especially the lateral tilting moment at the top of the tower, thus causing a larger impact on the dynamical behavior of the tower downwind and crosswind. It can be seen that, as the flexibility of the tower system increases, the interaction between the tower vibration and the pneumatic load also increases gradually. Taking the influence of the flexible tower on the aeroelastic load of the wind turbine into account can help predict the wind turbine pneumatic load more accurately and improve the efficiency of wind energy utilization; on the other hand, it can analyze the dynamic behavior of the flexible structure of the wind turbine more accurately, which is extremely beneficial to the structural optimization design of the wind turbine. On the other hand, it can analyze the dynamic behavior of the flexible structure of wind turbines more accurately, which is extremely beneficial to the structural optimization design of wind turbines.

**Author Contributions:** Conceptualization, J.H. and Y.C.; methodology, J.H.; software, J.C.; validation, J.H., Z.W. and J.C.; formal analysis, J.H.; investigation, Z.W.; resources, W.Y.; data curation, W.Y.; writing—original draft preparation, J.H.; writing—review and editing, J.H.; visualization, Z.W.; supervision, Y.C.; project administration, J.C.; funding acquisition, Y.C. All authors have read and agreed to the published version of the manuscript.

**Funding:** This research was funded by Science and Technology Planning Project of Guangdong Province grant number 2015B020240003, the National Science Foundation of China, grant number 51976113.

**Conflicts of Interest:** The authors declare no conflict of interest.

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
