# Peer review of "Influence of the Flexible Tower on Aeroelastic Loads of the Wind Turbine"

_applsci, doi:10.3390/app11198876_

Round 1

Reviewer 1 Report

Thank you for possibility of review of the paper. The paper is interesting as the effect of complex model of wind turbine, where the multi-factors were considered.

The model of wind turbine is very general, as the effect of constant mass consideration, is disadvantage of calculated model, but including the wind influence on the wing rotation (blade position) is strong advantage – only neglected dumping and free vibration (only my wish). The mathematical model (equation) is good presented and described.

Presented analysis results are good and in my opinion beneficial for further work on floating wind towers.

I recommend to publish paper in present form.

Author Response

Thank you for your letter and for the reviewers’ comments concerning our manuscript entitled ‘Influence of the flexible tower on aeroelastic loads of the wind turbine’ (ID: applsci-1366155). Those comments are all valuable and very helpful for revising and improving our paper, as well as the important guiding significance to our researches. 

Due to time issues, we were unable to do so to add to this manuscript, and we will continue to improve the follow-up work.

Reviewer 2 Report

The manuscript presents a computational methodology to calculate the dynamic behavior of wind turbines. Particularly the effect of the tower flexibility is investigated in a coupled model in order to analyze the effect of the motion of the system in the aerodynamic loads.

The paper would need a significant change in order to be accepted. The actual contribution of the presented study is difficult to extract from the manuscript while some well-known concepts are over-explained. The proposed model seems as well too simplified compared to current and nowadays models for wind turbines.

In the introduction section there is not an in depth study of the state of the art in the field. It would be interesting that the authors should provide a sufficient review of the current developments in the area acknowledging previous and recent works. Many aspects discussed in the paper are already known in the literature.

  • Visser. The aeroelastic code FLEXLAST. In B. Maribo Pedersen, editor, State of the Art of Aeroelastic Codes for Wind Turbine Calculations, Lyngby, Denmark, 1996
  • Rasmussen and M. H. Hansen. Present Status of Aeroelasticity of Wind Turbines. Wind Energy, 6:213–228, 2003.
  • C. Quarton. Wind turbien design calculations the state of the art. In A. Zervos, H. Ehmann, and P. Helm, editors, European union wind energy conference 1996, pages 10–15, Goteborg, Sweden, 1996
  • Bing Feng Ng, Rafael Palacios, Eric C. Kerrigan, J. Michael R. Graham, Henrik Hesse: Aerodynamic load control in horizontal axis wind turbines with combined aeroelastic tailoring and trailing-edge flaps.
  • Zhanwei LI, Binrong Wen, Xingjian Dong, Zhike Peng, Yegao Qu, Wenming Zhang: Aerodynamic and aeroelastic characteristics of flexible wind turbine blades under periodic unsteady inflows.
  • Aditya K. Sabale, Nagendra K. V. Gopal, Nonlinear Aeroelastic Analysis of Large Wind Turbines Under Turbulent Wind Conditions, AIAA Journal, 10.2514/1.J057404, (1-17), (2019).
  • Donghoon, D. H. Hodges, and M. J. Patil. Multi-flexible-body Dynamic Analysis of Horizontal Axis Wind Turbines. Wind Energy, 5:281–300, 2002

There are several terms that are no the common standard. The authors use the term “attitude” many times on the manuscript. It is not clear what the authors are referring to. In line 40 of page 1 the Blade Element Momentum theory (BEM) is named as Leaf Element Momentum theory. In line 133 of page 6, “order” is used to name what is commonly known as “mode” or “shape” of vibration. The word order is also used to refer to the fundamental or natural frequencies of the system. The common notation on this is to numerate them as first, second, third, and so on, frequencies and modes of vibration.

In section 1.3 authors discuss the effects of the soil-structure interaction upon the natural frequencies of the turbine and conclude that: “the flexible foundation of the tower has an important effect on the free vibration of the tower”. This is already studied in the literature:

  • Harte, M. & Basu, Biswajit & Nielsen, S.R.K.. (2012). Dynamic analysis of wind turbines including soil-structure interaction. Engineering Structures. 45. 509–518. 10.1016/j.engstruct.2012.06.041.

The manuscript is excessively thorough in well-known topics and developments, for example the use of transformation matrix to changes in the coordinate axes. The calculation method again with excessive explanations on the coordinate transformations, the time integration via the Newmark method while the main contribution of the study should be put in first plane for the reader.

Little reference is made on how the loads are calculated and applied in the model and for which wind conditions. Examples used in the manuscript should be completely described in order to make possible their reproduction by the readers or other researchers.

Reviewer 3 Report

The dynamic behavior of the blades and the tower during the operation of the wind turbine is an important topic.

The article is structured excellently and the results are given very well. The conclusions are adequate.
I would recommend that more literature sources be cited if possible.

Author Response

Dear Editors and Reviewers:

Thank you for your letter and for the reviewers’ comments concerning our manuscript entitled ‘Influence of the flexible tower on aeroelastic loads of the wind turbine’ (ID: applsci-1366155). Those comments are all valuable and very helpful for revising and improving our paper, as well as the important guiding significance to our researches. We have studied comments carefully and have made correction which we hope meet with approval. Revised portion are marked in red in the paper. 

We add a new reference [15] to modal frequencies on page 5, line 117 of the manuscript to facilitate better learning by the reader.

We add a new reference [16] to section of the blade on page 7, line 169 of the manuscript to facilitate better learning by the reader.

We add a new reference [17] to numerical examples on page 13, line 260 of the manuscript to facilitate better learning by the reader.

Regarding some inappropriate English language and styles, we have re-edited, especially for the summary and conclusion sections of the manuscript.

Round 2

Reviewer 2 Report

Authors have improved the manuscript.